# Guaianolide Sesquiterpene Lactones from *Centaurothamnus maximus*

**DOI:** 10.3390/molecules26072055

**Published:** 2021-04-03

**Authors:** Taha A. Hussien, Tarik A. Mohamed, Abdelsamed I. Elshamy, Mahmoud F. Moustafa, Hesham R. El-Seedi, Paul W. Pare, Mohamed-Elamir F. Hegazy

**Affiliations:** 1Pharmacognosy Department, Faculty of Pharmacy, Sphinx University, New Assiut City, Assiut 10, Egypt; thussien71@yahoo.com; 2National Research Centre, Chemistry of Medicinal Plants Department, 33 El-Bohouth St., Dokki, Giza 12622, Egypt; ta.mourad@nrc.sci.eg; 3National Research Centre, Department of Natural Compounds Chemistry, 33 El-Bohouth St., Dokki, Giza 12622, Egypt; ai.el-shamy@nrc.sci.eg; 4Department of Biology, College of Science, King Khalid University, Abha 61421, Saudi Arabia; mfmostfa@kku.edu.sa; 5Department of Botany & Microbiology, Faculty of Science, South Valley University, Qena 83523, Egypt; 6Department of Molecular Biosciences, The Wenner-Gren Institute, Stockholm University, S-106 91 Stockholm, Sweden; 7International Research Center for Food Nutrition and Safety, Jiangsu University, Zhenjiang 212013, China; 8Department of Chemistry, Faculty of Science, Menoufia University, Shebin El-Kom 32512, Egypt; 9Department of Chemistry & Biochemistry, Texas Tech University, Lubbock, TX 79409, USA; paul.pare@ttu.edu; 10Department of Pharmaceutical Biology, Institute of Pharmaceutical and Biomedical Sciences, Johannes Gutenberg University, Staudinger Weg 5, 55128 Mainz, Germany

**Keywords:** *Centaurothamnus maximus*, Asteraceae, guaianolides, flavonoids, biosynthesis, chemotaxonomy

## Abstract

*Centaurothamnus maximus* (family Asteraceae), is a leafy shrub indigenous to the southwestern Arabian Peninsula. With a paucity of phytochemical data on this species, we set out to chemically characterize the plant. From the aerial parts, two newly identified guaianolides were isolated: 3β-hydroxy-4α(acetoxy)-4β(hydroxymethyl)-8α-(4-hydroxy methacrylate)-1α*H*,5α*H*, 6α*H*-gual-10(14),11(13)-dien-6,12-olide (**1**) and 15-descarboxy picrolide A (**2**). Seven previously reported compounds were also isolated: 3β, 4α, 8α-trihydroxy-4-(hydroxymethyl)-lα*H*, 5α*H*, 6β*H*, 7α*H*-guai-10(14),11(13)-dien-6,12-olide (**3**), chlorohyssopifolin B (**4**), cynaropikrin (**5**), hydroxyjanerin (**6**), chlorojanerin (**7**), isorhamnetin (**8**), and quercetagetin-3,6-dimethyl ether-4’-*O*-β-d-pyranoglucoside (**9**). Chemical structures were elucidated using spectroscopic techniques, including High Resolution Fast Atom Bombardment Mass Spectrometry (HR-FAB-MS), 1D NMR; ^1^H, ^13^C NMR, Distortionless Enhancement by Polarization Transfer (DEPT), and 2D NMR (^1^H-^1^H COSY, HMQC, HMBC) analyses. In addition, a biosynthetic pathway for compounds **1**–**9** is proposed. The chemotaxonomic significance of the reported sesquiterpenoids and flavonoids considering reports from other *Centaurea* species is examined.

## 1. Introduction

*Centaurothamnus maximus* Wagentz and Dittri (Asteraceae) is a branched shrub that grows to a height of ca. 1.5 m [1]. Stems are densely white-tomentose with alternating leaves that are lanceolate to elliptic (2–6 cm wide by 8–15 cm long), silvery below and green above. Thistle-like magenta flowers 3–4 cm long at the end of the branches have a faint sweet scent [2]. *Centaurothamnus* is a monotypic genus from *Centaurea* and endemic to the mountains of the southwestern Arabian Peninsula. *C. maximus* (Forssk.) was first reported in 1775 from a collection from Yemen [1,2,3]. The genus is highly restricted in Saudi Arabia to cliffs and steep hillsides and is represented by ca. 200 species [4]. *C. maximus* is a paleoendemic species that presently grows in Yemen without any known traditional uses. This may be in part due to the plant’s limited distribution to high mountains cliffs and slopes in Yemen [3].

Previous *Centaurothamnus* phytochemical studies have led to the isolation of the sesquiterpene lactones guaianolides, edusamanolides, germacranolides, and elemanolides [5,6,7,8,9], as well as several flavonoids [10] and acetylenes [11,12]. Reports concerning the phytoconstituents of *C. maximus* include the guaianolide sesquiterpene lactones chlorojanerin, janerin, and cynaropicrin [13], as well as an oxygenated homoditerpenoid [14] and an aliphatic ester 8′α-hydroxy–*n*-decanyl-*n*-docosanonate [15].

In the current investigation, we describe the isolation and identification of two new guaianolide sesquiterpene lactones, 3β-hydroxy-4α(acetoxy)-4β(hydroxymethyl)-8α-(4-hydroxy methacrylate)-1α*H*,5α*H*,6α*H*-gual-10(14),11(13)-dien-6,12-olide (**1**) and 15-descarboxy picrolide A (**2**), as well as seven known compounds, 3β,4α,8 α-trihydroxy-4-(hydroxymethyl)-lα*H*,5α*H*,6β*H*,7α*H*-guai-10(14),11(13)-dien-6,12-olide (**3**), chlorohyssopifolin B (**4**) [16], cynaropicrin (**5**) [17], hydroxyjanerin (**6**) [18], chlorojanerin (**7**) [19], isorhamnetin (**8**) [20], and quercetagetin-3,6-dimethyl ether-4’-*O*-β-d-pyranoglucoside (**9**) [21] (Figure 1). In addition, biosynthetic pathways for the secondary metabolites (**1–9**) and the chemotaxonomic significance of sesquiterpene lactones and flavonoids are discussed.

## 2. Results

### 2.1. Structure Elucidation of the Isolated Compounds

A CH_2_Cl_2_:MeOH (1:1) of *C. maximus* aerial parts total extract was chromatography fractionated and purified, leading to two guaianolide sesquiterpene lactones: 3β-hydroxy-4α(acetoxy)-4β(hydroxymethyl)-8α-(4-hydroxy methacrylate)-1α*H*,5α*H*,6α*H*-gual-10(14),11(13)-dien-6,12-olide (**1**) and 15-descarboxy picrolide A (**2**) (Figure 1).

Compound **1**, a white amorphous powder, showed a molecular ion peak [M + H]^+^ at *m*/*z* 423.1655 (calcd. for C_21_H_27_O_9_, 423.1662), confirmed by high-resolution FAB-MS analysis, and an optical rotation of [α]D 25 = +17.0 (c, 0.001, MeOH). Twenty-one carbons were detected through the ^13^C NMR spectrum (Table 1), which was incompatible with its molecular formula. The classification of these carbons was inferred from the DEPT analyses as; one methyl, seven methylenes (three olefinic), six methines (three oxygenated at δ_C_ 76.1, 77.1, and 74.2), and seven quaternary carbons (three olefinic and three keto at δ_C_ 169.5, 171.9, and 156.2 (Table 1). ^1^H NMR analysis (Table 1) showed a characteristic large coupling pattern of oxymethine proton at δ_H_ 4.84 (^1^H, t*, J*_5,6_ = 12.9), assigned to a lactone proton at C-6 that specified to a trans-diaxial character of the protons for C-5 (δ_H_ 2.36, t, *J* = 9.9) and C-7 (δ_H_ 3.22, brt, *J* = 9.3), strongly suggesting a guaiane-type sesquiterpene lactone skeleton [22]. With the exception of acetoxy group at C-4 (δ_C_ 83.4) and chlorine atom with up-field chemical shift of C-15 (δ_C_ 63.4), both ^1^H and ^13^C NMR spectra for **1** were quite similar to those for compound **5**: (3β,4α-dihydroxy-4β-(hydroxymethyl)-8α-(4-hydroxy-methacrylate) l*αH*,5*αH*,6*βH*,7*αH*-guai-10(14),11(14)-dien-6,12-olide), which has been previously isolated from *Amberboa ramosa* [16].

The appearance of a sharp singlet signal at δ_H_ 2.11 (3H, s, H-1) together with new ester carbonyl at δ_C_ 171.9 in **1**, along with an absence of these signals in **5**, indicates the presence of an acetoxy group instead of hydroxyl group at C-4. While it is possible that such an acetate functionality is a chemical artifact (e.g., drying and/or extraction), the fact that such derivatizations have been phytochemically studied in the same manner from the same genus and other genera suggests that these natural products are in fact plant-derived metabolites.

Two-dimensional NMR data (^1^H-^1^H COSY, HMQC and HMBC) clearly indicate that the acetoxy of **1** is comparable to that of **5 [16]**. HMBC correlations (Figure 2) were observed between H-8/C-16 and H-18/C-16, C-17, and C-19, supporting the sequence and position of the side chain at C-8. In addition to two- and three-bond correlations between H-1/C-3, C-6, C-7, C-10; H-2/C-3, C-4, C-5, C-6, C-10, C-14; H-5/C-1, C-2; H-6/C-1, C-8; H-7/C-1; H-14/C-1, C-8, C-9; and H-8/C-6, C-14 were further confirmation of the structure of **1** (Figure 2).

The relative stereochemistry of **1** was determined by comparison of the coupling constants and the biogenetic correlation with the corresponding guaianolides isolated from other Asteraceae species [23,24]. NOESY correlations (Figure 3) between H-1, H-3, H-5, and H-9 are in an α orientation, and NOESY correlations between H-6, H-8, and H-15 are in a β orientation. From the spectral data reported here, **1** was identified as 3β-hydroxy-4α(acetyloxy)-4β(hydroxymethyl)-8α-(4-hydroxy methacrylate)-1α*H*,5α*H*,6α*H*-gual-10(14),11(13)-dien-6,12-olide, a newly reported natural product.

Compound 2, a yellowish amorphous powder, had an optical rotation of [α]D 25 = +7.0 (c, 0.005, MeOH). A base peak at *m*/*z* 472.1758 [M] corresponded to a molecular formula of C_25_H_28_O_9_ (calcd. for C_25_H_28_O_9_, 472.2042) in the HR-FAB-MS spectrum. ^13^C NMR and DEPT spectral analyses (Table 1) revealed 25 carbons that were classified as seven methylenes (three olefinic), 10 methines (three oxygenated at δ_C_ 76.1, 77.2, and 74.2), and eight quaternary carbons (three olefinic as well as two keto at δ_C_ 165.5 (C-12) and 169.5 (C-16), (Table 1). ^1^H NMR spectral data (Table 1) show characteristic resonances for a*p*-disubstituted benzene moiety at δ_H_ (2H, *J* = 10.5, H-2′,6′) and 6.85 (2H, *J* = 2.4, H-3′,5′). Aromatic carbons were observed at δ_C_ 138.1, 114.6, 131.7, and 124.5, confirming *p*-disubstituted aromatic moiety. Proton signals were similar to a previously reported picrolide A isolated from *Acroptilon repens* [25]. Signal alternations included an ester carbonyl carbon for picrolide A reported at δ_C_ 167.1, which was missing in **2**, suggesting that the aromatic ring attached at C-15 and present as *p*-dihydroxy benzene moiety in **2** was modified to a*p*-hydroxy benzoate moiety in picrolide A. The HMBC connections (Figure 2) from H-15 to C-1′/C-3/C-4; H-2′,6′ to C-3′/C-4′/C-5′; and H-3′,4′ to C-1′/2′/6′ further confirmed the structure and location of the aromatic moiety at C-15. Three correlations were also observed from H-8 to C-16, H-18 to C-16/C-17, and H-19 to C-16/C-17/C-18, supporting the sequence and position of the side chain at C-8. In addition, two- and three-bond correlations from H-1 to C-3/C-6/C-7/C-10, H-2 to C-3/C-4/C-5/C-6/C-10/C-14, H-5 to C-1/C-2, H-6 to C-1/C-8, H-7 to C-1, H-14 to C-1/C-8/C-9, and H-8 to C-6/C-14 were further confirmation of the structure of **2** (Figure 2).

The relative configurations of both **1** and **2** were the same when compared with the corresponding guaianolides isolated from Asteraceae [23,24]. NOESY correlations (Figure 3) between H-1, H-3, H-5, and H-9 are in an α orientation, and NOESY correlations between H-6, H-8, and H-15 are in a β orientation. Accordingly, the structure of **2** was established as a new derivative of picrolide A and named 15-descarboxy picrolide A.

### 2.2. Proposed Biosynthetic Pathway of the Isolated Compounds

Generally, the terpenoids biosynthesis in plants can arise in dissimilar subcellular compartments, the cytosol, mitochondria, and/or plastids [26,27]. Biosynthetically, farnesyl diphosphate (FPP) is considered the main precursor for biosynthesis of a vast array of sesquiterpene. Cyclization of FPP into (+) germacrene A is catalyzed by (+) germacrene A synthase (GAS) [28]. The latter is converted to the corresponding acid, germacrene A acid, through hydroxylation and oxidation reactions catalyzed by cytochrome P_450_ germacrene A oxidase (GAO). Germacrene A acid is then hydroxylated at C-6 to produce 6-hydroxy-germacrene A acid as an unstable intermediate by the action of another cytochrome P_450_(+) costunolide synthase (COS). Costunolide is obtained from this intermediate that undergoes spontaneous non-enzymatic lactonization of the hydroxyl group at C-6 with the carboxylic group at C-12 (Figure 4) [29,30,31]. Costunolide is considered a branching point precursor for producing germacranolides, eudesmanolides, and guaianolides as the three major sesquiterpene lactones groups. Thus, 4,5 epoxidation of costunoilde is hypothesized to be the first committed step in guaianolide biosynthesis through the conversion of costunolide to parthenolidecatalyzed by parthenolide synthase (TpPTS) [31]. The opening of the epoxide through an intramolecular attack of the double bond generates the three-cyclic skeleton as a guaianolide-type intermediate that is responsible for generating a large variety of guaianolides (Figure 4) [29,32,33]. The guaianolides in Asteraceae have a specific biosynthetic pathway with unique conformations that differ from guaianolides in the family Apiaceae. The lactone ring in Apiaceae is either 6β, 8α or 6β, 8β, whereas in Asteraceae, it has only been seen as 6α, 8β [23,24]. The hydroxylase enzymes activate an enzymatic hydroxylation of the guaianolide-type intermediate at C-3, C-8, and C-15, thereby producing compound **3**. Compound **4** was biosynthetically proposed by incorporation of chloride atoms at C-15 of compound **3,** which catalyzed by Flavin adenine dinucleotide (FADH_2_)-dependent halogenases as the type of halogenating enzymes of compounds activated for electrophilic attack (Figure 4). On the other hand, the C-8 position is easily hydroxylated by the enzyme CYP71BL1 and acts as an active site to accept acyloxy moiety via the P_450_ acetyltransferase enzyme [34]. The generation of the side chain at C-8 is proposed via esterification of the hydroxyl group at C-8 with acrylic moiety, followed by methylation and hydroxylation of the side chain to generate compound **6**. The latter is considered the main precursor of compounds **1**, **2**, **5*,*** and **4** through specific biochemical pathways (Figure 4). Compound **7** is similar to 4 in chlorination of 6 at C-15, which is activated by FADH2-dependent halogenases. The acetylation of 6 at C-4 is believed to be mediated by an acetyl transferase, which catalyzes the transfer of an acetyl group from acetyl-CoA as a donor molecule to produce an acetylated analogue **1**. Compound **2**, however, may be obtained via condensation reaction of compound **6** at C-15 with a simple aromatic moiety, such as hydroquinone, which may be produced from shikimic acid as precursor. The dehydrogenase enzyme may be converting the primary alcoholic group at C-15 in compound **6** into a formyl group that also converted into a methyl one by reductase enzyme, and a double bond between C-4 and C-15 is formed by losing one molecule of water to produce compound **5** (Figure 4).

Flavonoids are products of a shikimic acid and the acetate pathway by condensation of 4-hydroxy cinnamoyl-coenzyme A, referred to as 4-coumaroyl coenzyme A, with unit 9 of malonyl coenzyme A. The plant utilizes a shikimic acid pathway for deriving a polyketide intermediate that forms a chalcone skeleton. The chalcone skeleton serves as a key intermediate in the biosynthesis of several classes of flavonoids [35,36,37]. The proposed biosynthetic pathway of compounds **8** and **9** is shown in Figure 5.

### 2.3. Chemosystematic Significance

*Centaurothamnus* is a monotypic genus that is a cross-taxon basionym with the genus *Centaurea.* The genus *Centaurea* is considered an attractive source for sesquiterpene lactones and flavonoids [38,39]. The *Centaurea* species are rich in various types of sesquiterpene lactones, including guaianolides, germacranolide, eudesmanolides, and elemanolides. The guaianolides are the most abundant and more distributed in genus *Centaurea* [40,41]. Based on bibliographic research, about 54 guaianolide sesquiterpene compounds have been isolated from 80 *Centaurea* species, as well as 20 germacronolides from 46 species, 7 elemanolides from 10 species and 4 eudesmanolides from 3 species [42]. There are also review articles that describe sesquiterpene lactones isolated from specific *Centaurea* species, including 20 guaianolides isolated from *C. acaulis*, *C. omphalotricha*, and *C. musimomum*; 11 germacranolides and 12 elemanolides from *C. acaulis*, *C. melitensis*, *C. calictrapa*, *C. foucauldiana*, *C. lippii*, *C. parviflora*, *C. tougourensis*, *C. sulphurea*, *C. papposa*, *C. sicula*, and *C.pullata;* and 8 eudesmanolides from *C. acaulis*, C*. papposa*, *C. granata*, *C. pullata* and *C. maroccana* [43]. In addition, guaianolides and elemanolides have been isolated from *Centaurea,* including four guaianolides and cynaratriol from *C. musimomum* [44]; centaurpensin and 13-acetyl solstitian A from *C. solstitialis* [45]; cebellin M from *C. bella* [46]; the elemanolideshierapolitanins A, B, C, and D isolated from *C. hierapolitana* [22]; and 13-*N*-proline melitensin and 13-*N*-proline-6α, 8α, 15-trihydroxy elema-1,3-diene-oic acid from *C. polyclada* [47].

Chlorinated guaianolides with noted medical value have been isolated from species of *Centaurea* [48]. The chlorinated guaianolide derivatives were reported as chlorohyssopifolins A and C, which were isolated from *C. bella*, *C. carthelinica*, *C. aegyptiaca*, *C. colchica*, *C. dealbata*, *C. exsurgens*, *C. hyssopifolia*, *C. janeri*, *C. karabaghensis*, *C. somchetica*, *C. taochia,* and *C. zangezuri* [49]; chlorohyssopifolin B reported from *C. aegyptiaca, C. Hyssopifolia* [49], and *C. linifolia* [50]; chlorohyssopifolins D and E from *C. hyssopifolia* [51] and *C. linifolia* [50]; chlorojanerinfrom *C. aegyptica* [52], *C. hyssopifolia* [53] and *C. sinaica* [54]; chlororepdiolide from *C. repens*; 19-desoxychlorojanerin from *C. aegyptiaca* [48]; elegin from *C. repens* [55] and *C. solistitialis* [56]; *epi*-centaurepensin, episolistiolide, and linichlorin A from *C. linifolia* [49]; solistitiolide from *C. repens* and *C. solistitialis* [56]; 14-chloro-10-β-hydroxy-10(14)-dihydrozaluzanin D from *C. acaulis* [57]; and cebellin C and centaurepensin from *C. bella* [58].

On the other hand, flavonoids are also distributed in different species of *Centaurea*. Different reports showed that 119 flavonoids of various types, primarily belonging to the flavone class, have been isolated from 53 *Centaurea* species. Apigenin and luteolin, as well as their glycosides, were the most common flavones in the *Centaurea* species. The glycosylation of flavones was commonly found at position 7 as 6-methoxykaempferol 7-*O*-glucoside, kaempferol 7-*O*-glucoside, along with hispidulin-7-*O*-glucoside from *C. macrocarpa* [59], apigenin 4′-(6′-methylglucuronide) from *C. nicaeensis* [60], and apigetrin from *C. resupinate* [61]. The flavonol glycosides derivatives were rare; only patuletin-7-*O*-glucoside was isolated from *C. macrocarpa* [59] and nicotiflorin as a flavonol-3-*O*-glucoside from *C. resupinate* [61].

The methoxylated flavone derivatives were commonly detected in genus *Centaurea* as 6-mono-methoxyflavones, which are represented as 6-methoxykaempferol and 6-methoxyluteoin from *C. macrocarpa* [59], 5,7-dihydroxy-6-methoxyflavone (oroxylin A), and 5,7,4′-trihydroxy-6-methoxyflavone (hispidulin) from *C. ragusina* [62]. Other rare methoxylated flavone derivatives include 3′-mono-methoxyflavones, such as chrysoeriol from *C. resupinate* [61] and 6,5′-dimethoxyflavones such as jaceosidin from *C. nicaeensis* [60]. 6,7,3′-Trimethoxyflavone was found as cirsilineol from *C. nicaeensis* [60]. The 5,7-dihydroxyflavone derivative chrysin was reported from both *C. ragusina* [62] and *C. resupinata* [61], and the 5,7, 4′-trihydroxyflavone derivative apigenin was isolated from *C. resupinate* [61]. Prunasin, a cynogenic glycoside, was observed in *C. nicaeensis* [60].

In the present study, seven guaianolide sesquiterpene, including two chlorinated guaianolides, namely chlorohyssopifolin B (**4**) [16] and chlorojanerin (**7**) [19], together with two flavonoid compounds, 5,7,4′-trihydroxy-3,6-dimethoxyflavone-4′-*O*-glucoside (9) and isorhamnetin (8) (Litvinenko and Bubenchikova, 1988), were isolated from *C. maximus*. Compound **3** was firstly isolated from genus *Centaurea* and previously characterized from *Amberboaramosa* [16]. Chlorohyssopifolin B (**4**) was also previously reported from *Amberboaramosa* [16], *C. aegyptiaca* [48], *C. maximus*, *C. hyssopifolia,* and *C. linifolia* [49]; **5** was previously isolated from *C. amberboa*, *C. dealbata*, *C. exarate* [49], *C. linifolia,* and *C. maximus* [63]; **6** was firstly isolated from *Centaurea* and previously isolated from *Amberboaramosa* [16] and *Cousiniaaitchisonii* [18]; and **7** (chlorojanerin) was previously reported from *C. adjarica* [64], *C. Janeri* [53], *C. sinaica* [54], and *C. Aegyptiaca* [48]. Compounds **1** and **2** were isolated and characterized as new guaianolides from *C. maximus*. The 3′ methoxyflavonol derivative **8** was first characterized from *C. maximus* and previously isolated from *C. kotschyi* [65], *C. cynaus* [20], and *C. aegyptiaca* [66], whereas 5,7,4′-trihydroxy-3,6-dimethoxyflavone-3′-*O*-glucoside (**9**) was isolated as a new natural compound from *C. maximus* [21].

Based on the above studies, it appears that guaianolide sesquiterpene lactones isolated from *C. maximus* are similar to those reported from *C. aegyptiaca*, and *C. linifolia,* which have overlapping biosynthetic pathways and are characterized by their potential to produce chlorinated guaianolides as well as methoxylated flavonoid derivatives.

## 3. Materials and Methods

### 3.1. General Procedures

Optical rotations were recorded on a JASCO P-2300 polarimeter (Jasco Corporation, Tokyo, Japan). NMR and HR-MS spectra were recorded on a Bruker 500 NMR (Bruker, Billerica, MA, USA) and JEOL JMS-700 (JEOL, Ltd., Tokyo, Japan) instrument, respectively. A JASCO 810 spectropolarimeter was used for circular dichroism.

Chromatographic separation was applied using semi-preparative Agilent 1200 High performance liquid chromatography (HPLC) with a refractive index (RI) detector (Santa Clara, CA, USA) and reversed-phase column chromatography (YMC-Pack™ octadecylsilyl (ODS) column (250 × 10 mm, 5 μm), Marcon Boulevard, Allentown, PA, USA). Silica gel 60 (Merck, 230–400 mesh; Merck, Darmstadt, Germany) and precoated RP–18 F_254_ plates (Merck, Darmstadt, Germany) were used for column chromatography and Thin Layer Chromatography (TLC) analysis, respectively.

### 3.2. Plant Material

The wild aerial parts of *C. maximus* were collected in March 2015 from Al Udayn, Ibb, Yemen. The plant was kindly identified by Prof. Dr. Abdulnaser Al Gifri of the Biology Department at Education College, Aden University, Yemen. A voucher specimen (P 610) was deposited in the Pharmacy Department at the University of Sciences and Technology, Ibb, Yemen.

### 3.3. Extraction and Isolation

The air-dried powder of *C. maximus* (1 kg) was extracted using methylene chloride-methanol (1:1; *v*/*v*, 4 L) at room temperature. A gummy residue (90.5 g) extract was produced via in vacuo concentration, followed by fractionation on silica gel flash column chromatography using *n*-hexane (3 L, 100%) and then by gradient of *n*-hexane/ethyl acetate up to 100% ethyl acetate and ethyl acetate–methanol up to 15% MeOH (2 L of each solvent mixture). Twelve major fractions were collected together based on the TLC profile. Vanillin–sulfuric acid reagent was used for the detection of compound spots to yield the following six fractions: A (10.0 g), B (6.0 g), C (11.5 g), D (15.0 g), E (13.0 g), and F (7.5 g). Fraction C was subjected to further fractionation on the ODS column (3 × 60 cm); the eluted gradient with Elution started with 100% water, and the polarity was decreased with 10% increments of methanol until reaching 100%. The subfractions were obtained and subjected to isolation and purification by RP-18 HPLC (20 × 250 cm) using 80:20% MeOH:H_2_O, (8:1, 2.5 L) to produce compound **5** (20.0 mg) and **6** (15.0 mg). Fraction D was also subjected to further fractionation on the ODS column (3 × 60 cm) using 80:20% (MeOH:H_2_O) and finally eluted with 100% MeOH. The obtained fractions were further purified by RP-18 HPLC using MeOH:H_2_O (7:2, 2.5 L) with a flow rate of 5 mL/min to obtain **1** (11.5 mg), **2** (13.5 mg), and **9** (10 mg). Fraction E was purified by RP-18 HPLC using MeOH (6:3, 2.5 L) with a flowrate of 6 mL/min to produce **3** (7.5 mg) and **4** (12 mg). Fraction F was purified by RP-18 HPLC using MeOH:H_2_O (50:50%, 2.5 L) with a flowrate of 5 mL/min to afford **7** (9.5 mg) and **8** (15.0 mg). NMR assignments for isolated secondary metabolites (**1**-**9**) were included in the Appendix A.

*3β-Hydroxy-4α(acetoxy)-4β(hydroxymethyl)-8α-(4-hydroxymethacrylate)-1αH, 5αH, 6αH-gual-10(14), 11(13)-dien-6,12-olide* (**1**). Amorphous powder; [α]D 25 = +17.0 (c, 0.001, MeOH); ^1^H (CDCl_3_, 500 MHz) and ^13^C (CDCl_3_, 125 MHz) NMR; see Table 1. FAB-MS *m*/*z* = 423.1655 [M + H]^+^; HR-FAB-MS *m*/*z* = 423.1655 (calcd. for C_21_H_27_O_9_, 423.2222).

*15-Descarboxy picrolide A* (2). Amorphous powder; [α]D 25 = +7.0 (c, 0.005, MeOH); ^1^H (CDCl_3_, 500 MHz) and ^13^C (CDCl_3_, 125 MHz) NMR; see Table 1. FAB-MS *m*/*z* = 472.1758 [M]^+^; HR-FAB-MS *m*/*z* = 472.1758 (calcd. for C_25_H_28_O_9_, 472.2042).

## 4. Conclusions

Two new guaianolide sesquiterpene lactones (**1**–**2**) were characterized from *C. maximus,* as well as five known guaianolide sesquiterpene (**3**–**7**), including two chlorinated guaianolides (**4** and **7**), together with two known flavonoids (**8**–**9**). Compounds **3**, **6** and **9** were firstly isolated from genus *Centaurothamnus*, while, **4** and **8** were isolated for the first time from *C. maximus*. Biosynthetically, costunolide is considered the branching point precursor for producing germacranolides, eudesmanolides, and guaianolides, however, 4,5 epoxidation of costunoilde is hypothesized to be the first committed step in the Asteraceae family for guaianolide biosynthesis through the conversion of costunolide to parthenolide. By opening of the epoxide through an intramolecular attack of the double bond, the three-cyclic skeleton is generated as a guaianolide type intermediate and can go on to generate a variety of enzymatically mediated guaianolides. Guaianolide biosynthesized in the family Asteraceae have a specific biosynthetic pathway with the lactone ring having a 6α, 8β conformation. Based on chemosystematic analysis, guaianolide sesquiterpenes from *C. maximus* exhibit chemical overlap with *Centaurea aegyptiaca*, and *C. linifolia*, confirming their placement in one section. Additional data on guaianolide sesquiterpenes and flavonoids from other *Centaurea* species will be required to further elucidate intergeneric relationships.

## Figures and Tables

**Figure 1 molecules-26-02055-f001:**
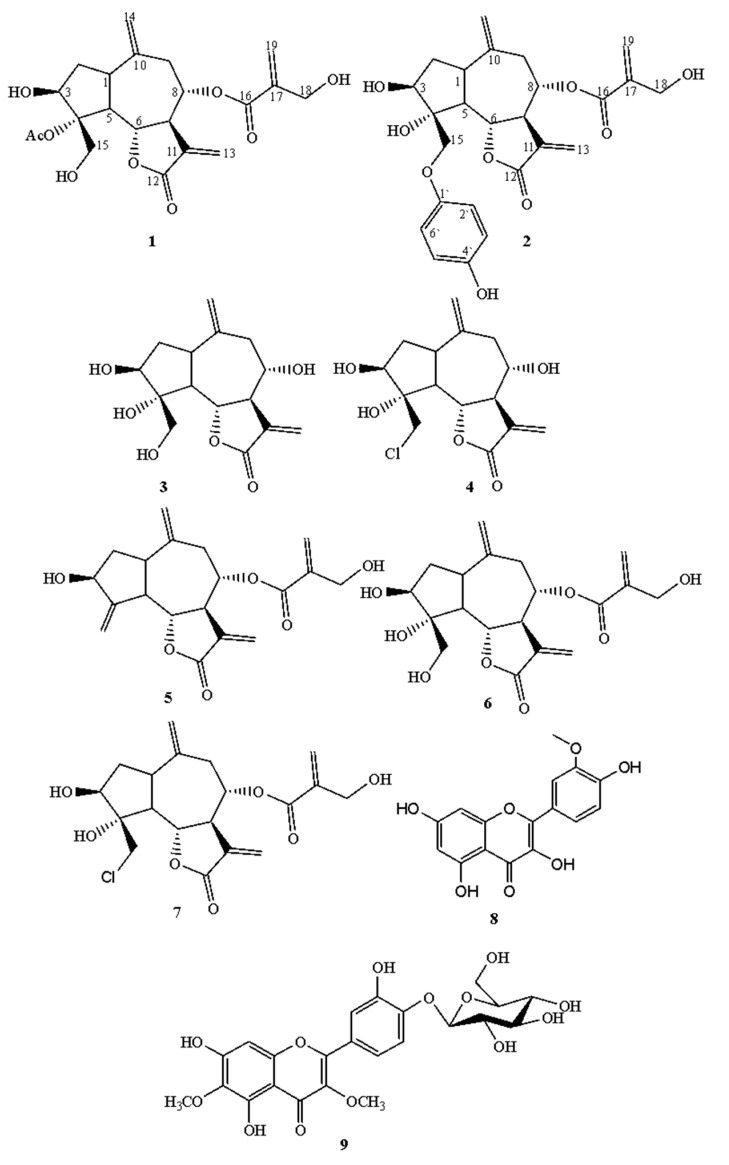
Structures of the isolated compounds from *Centaurothamnus maximus.*

**Figure 2 molecules-26-02055-f002:**
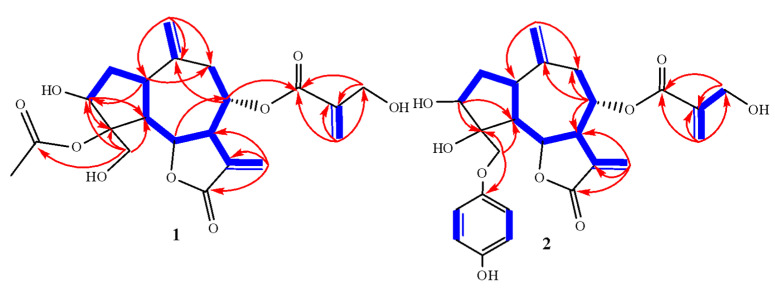
Observed ^1^H-^1^H-COSY (bold blue line) and HMBC (red arrow) correlations for **1** and **2.**

**Figure 3 molecules-26-02055-f003:**
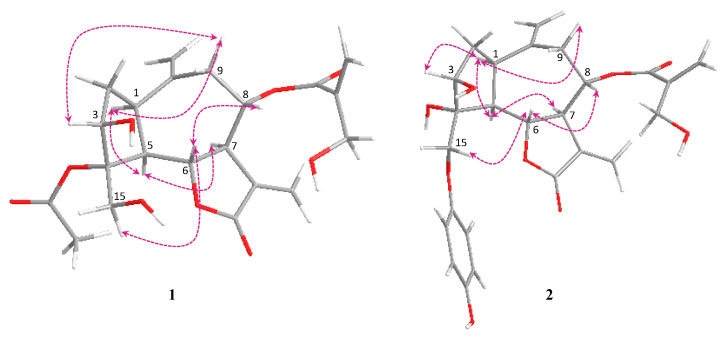
Significant NOESY correlations of **1** and **2.**

**Figure 4 molecules-26-02055-f004:**
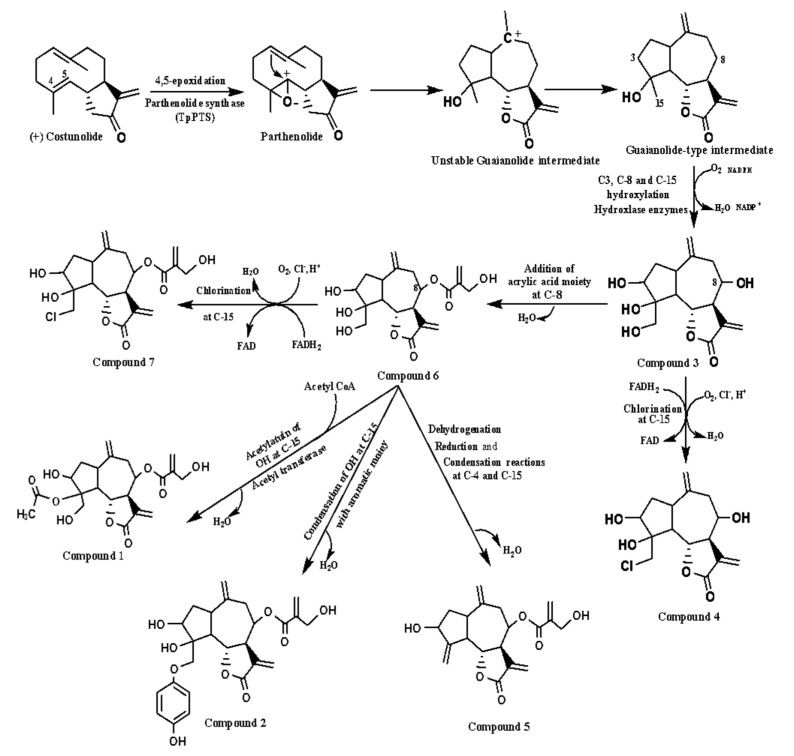
Biosynthesis proposed pathway scheme for the isolated guaianolides.

**Figure 5 molecules-26-02055-f005:**
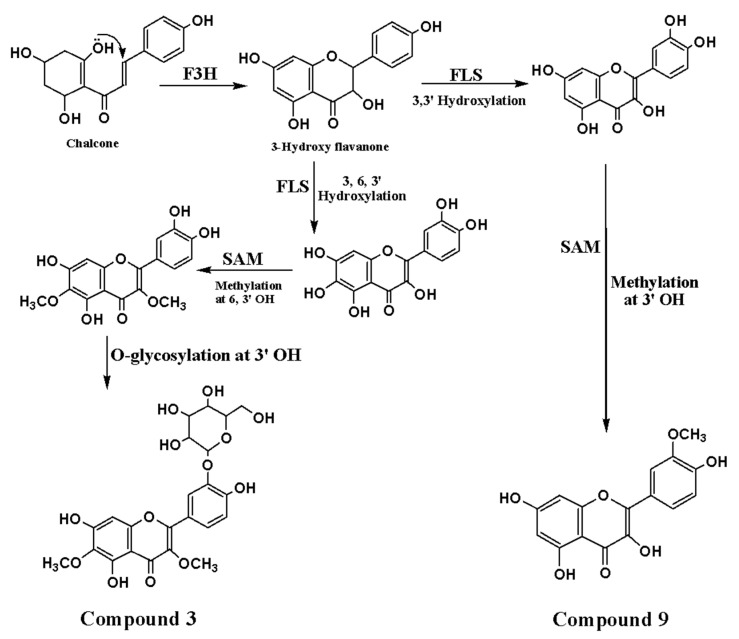
Proposed biosynthesis pathway scheme for the isolated methoxyled flavones.

**Table 1 molecules-26-02055-t001:** ^1^H and ^13^C NMR data of compounds **1**–**7** in CDCl_3_ (500 and 125 MHz *δ* in ppm, *J* in Hz).

No.	1		2		3	4	5	6	7
	*δ* _H_	*δ* _C_	*δ* _H_	*δ* _C_	*δ* _C_	*δ* _C_	*δ* _C_	*δ* _C_	*δ* _C_
**1**	3.52 m *	46.4	3.60 m *	46.3	43.4	47.1	44.8	44.5	46.1
**2**	1.60 m *, 2.35 m *	38.3	1.60 m *	39.8	43.3	38.6	38.6	37.6	38.6
**3**	4.05 dd (7.5, 5.7)	76.1	4.16 m *	76.1	77.5	75.6	72.8	77.0	75.6
**4**	-	83.4	-	83.8	83.1	84.4	152.6	83.9	84.5
**5**	2.36 t (9.9)	57.1	2.45 dd (17.2, 6.5)	57.3	55.1	58.4	50.7	55.6	58.3
**6**	4.84 t (12.9)	77.1	4.90 (21.5)	77.2	77.8	77.5	78.9	77.4	77.1
**7**	3.22 brt (9.3)	48.2	3.35 m*	47.3	51.2	49.2	48.2	47.4	47.8
**8**	5.15 ddd (10.5, 4.2, 2.0)	74.2	5.16 m	74.2	71.9	71.1	74.3	74.4	74.1
**9**	2.75 dd (13.5, 5.9), 2.42 m	35.0	2.76 dd (7.5, 6.9), 2.39 dd (12.9,10.7)	38.4	36.5	38.6	36.3	37.0	34.4
**10**	-	143.0	-	143.5	142.8	143.8	142.6	142.1	143.6
**11**	-	138.0	-	140.6	137.3	139.0	138.3	136.8	138.0
**12**	-	169.5	-	165.2	170.2	170.2	169.9	169.9	169.5
**13**	6.10 d (3.8), 5.60 d (3.24)	120.8	5.66 d (4.1), 6.12 d (4.1)	121.5	124.1	121.2	121.1	123.3	120.8
**14**	5.14 brs, 4.82 brs	116.5	5.1 brs, 4.75 brs	117.0	116.1	115.5	116.8	117.4	116.1
**15**	4.50 d (12.0)	66.5	4.59 d (13.9)	66.2	63.6	48.7	111.5	63.4	48.7
**16**	-	165.2	3.4 brs, 2.41 brs	169.5			165.17	165.5	165.2
**17**	-	140.6	5.99 d (3.6), 6.33 d (3.6)	138.1	-	-	140.5	139.7	140.6
**18**	4.33 brs	60.2	4.33 s	60.2	-	-	60.3	61.4	60.2
**19**	6.32 brs, 5.99 brs	124.5	-	124.5	-	-	124.6	126.3	124.5
**1′**	-		-	138.1	-	-	-	-	-
**2′**	-		6.85 d (10.5)	114.6	-	-	-	-	-
**3′**	-		8.00 d (10.5)	131.7	-	-	-	-	-
**4′**	-		-	124.5	-	-	-	-	-
**5′**	-		8.00 d (10.5)	131.7	-	-	-	-	-
**6′**	-		6.85 d (10.5)	114.6	-	-	-	-	-
**C=O, OAc**	-	171.9	-	-	-	-	-	-	-
**CH3, OAc**	2.11 s	19.5	-	-	-	-	-	-	-

* Overlapping signals. - Not detecetd.

## Data Availability

The data presented in this study are available in Appendix A.

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
