# Peer review of "Guaianolide Sesquiterpene Lactones from Centaurothamnus maximus"

_molecules, 2021, doi:10.3390/molecules26072055_

Round 1

Reviewer 1 Report

-Line 54: “We were interested in further probing the phytochemical constituents on this plant in search of possible bioacive metabolites.” For this manuscript, I suggest the authors remove the mention of searching for possibly bioactive compounds, since the authors did not get to test the isolated compounds for any biological activity at this point. the finding of two new compounds with differentiated structures and the proposition of their biosynthesis and chemosystematic relevance is already sufficient.

-List the positions in the structures according to the values ​​described in the NMR data table.

-Use hyphens for names, especially when indicating position or optical rotation (p.e. lines 220,223,225,…).

-Could the authors rule out the chance that acetylation at position 4 of compound 1 is a product of structural modification by using ethyl acetate in the purification procedure?

-Figure 4:  the second structure in figure 4 should be revised. The attack of the electron pair of the double bond is not adequately described, as well as the formation of the double bond of the side chain. add the full name of the enzymes involved in the biosynthesis described in figure 4. Since the authors propose to discuss biosynthesis, it would be interesting to comment on the possible origin of the aromatic system in structure 3.

-Line 161: Verify the name “crylic acid”

-Line 210: “Centaurea”: in italics

-Figure 5: the figure caption is incorrect.

-Line 299: “germacranoildes, eudsmanoides” misspelled.

Author Response

Response to Reviewer 1 Comments

Thanks for respected reviewer for his comments to improve our manuscript.

Point 1: -Line 54: “We were interested in further probing the phytochemical constituents on this plant in search of possible bioacive metabolites.” For this manuscript, I suggest the authors remove the mention of searching for possibly bioactive compounds, since the authors did not get to test the isolated compounds for any biological activity at this point. the finding of two new compounds with differentiated structures and the proposition of their biosynthesis and chemosystematic relevance is already sufficient.

Response 1: the sentence “searching for possibly bioactive compounds” have been removed.

Point 2: -List the positions in the structures according to the values ​​described in the NMR data table.

Response 2:the positions in the structures were added.

Point 3: Use hyphens for names, especially when indicating position or optical rotation (p.e. lines 220,223,225,…).

Response 3:hyphens for names were added

Point 4: Could the authors rule out the chance that acetylation at position 4 of compound 1 is a product of structural modification by using ethyl acetate in the purification procedure?

While the opportunity that such an acetate functionality is a chemical artifact (e.g. drying and/or extraction), the fact that such derivatized have been phytochemically studied in the same manner from the same genus or other genus suggests that these natural products are in fact plant-derived metabolites.

-Öksüz, S., Serin, S. and Topçu, G., 1994. Sesquiterpene lactones from Centaurea hermannii. Phytochemistry, 35(2), pp.435-438.

-Todorova, M.N., Ognyanov, I.V. and Shatar, S., 1991. Sesquiterpene lactones in mongolian Saussurea lipshitzii. Collection of Czechoslovak chemical communications, 56(5), pp.1106-1109.

-Singh, P. and Bhala, M., 1988. Guaianolides from Saussurea candicans. Phytochemistry, 27(4), pp.1203-1205.

Point 5:Figure 4:  the second structure in figure 4 should be revised. The attack of the electron pair of the double bond is not adequately described, as well as the formation of the double bond of the side chain. add the full name of the enzymes involved in the biosynthesis described in figure 4. Since the authors propose to discuss biosynthesis, it would be interesting to comment on the possible origin of the aromatic system in structure 3.

Response 5:According to the comment of reviewer 3, figure 4 was simplified to be started with costunolide as precursor. The full name of the enzymes involved in figure 4 were added. Also, the possible origin of aromatic moity in structure 2 were discussed.

 Point6: -Line 161: Verify the name “crylic acid”

Response 6: The name was verified and corrected to ‘’acrylic acid’’

Point 7: -Figure 5: the figure caption is incorrect.

Response 7: The caption of figure 5 was corrected.

Point 8: -Line 210: “Centaurea”: in italics

Response 8: Centaurea as a genus name was Italicized.

Point 9: -Figure 5: the figure caption is incorrect.

Response 9: Type mistake was corrected.

Point 10: Line 299: “germacranoildes, eudsmanoides” misspelled.

Response 10: Type mistake was corrected.

Reviewer 2 Report

The authors present the isolation from aerial parts of Centaurothamnus maximus, and the identification of two new two new guaianolides, 3β-hydroxy-4α(acetoxy)-4β(hydroxymethyl)-8α-(4-hydroxy methacrylate)-1αH, 5αH, 6αH-23 gual-10(14), 11(13)-dien-6,12-olide and 15-descarboxy picrolide A, as well as seven previously reported compounds.

In my opinion, the manuscript presents points to be considered:

1.- In line 80, it was mentioned that oxymethine protons with an δH 4.84 correspond to a lactone proton at C-7; however, these protons are located at C6 (Table 1).

2.- In line 81, the authors mentioned that the protons at C-5, C-6, and C-7 have a trans diaxial arrangement; however, the coupling constants of these protons’ signals support this asseveration were not provided. The authors must provide this information.

3.- In figure 3, NOE interactions between H3 – H9 and H5 H9 are shown; however, the orientation observed for the proton 9 in the conformers included in this figure do not become possible this interaction. Are the conformers presented in this figure the lowest energy ones? It is necessary the authors clarified this situation.

4.- The quality of figure 3 is not good. It is important to improve it.

5.- In the biosynthetic pathway discussion, the description of the compounds biotransformations to achieve product 6 is suitable, but the description of the biochemical pathways to produce compounds 1, 2, 5 and 4, actually is missing. The authors should include a brief description of these biotransformations.

In conclusion, the manuscript could be suitable for publication in Molecules if the authors attend the recommendations.

Author Response

Response to Reviewer 2 Comments

Thanks for respected reviewer for his comments to improve our manuscript.

Point 1: In line 80, it was mentioned that oxymethine protons with an δH 4.84 correspond to a lactone proton at C-7; however, these protons are located at C6 (Table 1).

Response 1: Type mistake of the assaigned carbon in the text was corrected to be C-6 as in Table 1.

Point 2: In line 81, the authors mentioned that the protons at C-5, C-6, and C-7 have a trans diaxial arrangement; however, the coupling constants of these protons’ signals support this asseveration were not provided. The authors must provide this information.

Response 2: Thanks for reviwer comment. The diaxial information regarding the C-5,C-6 and C-7 for suggesting a guaiane-type sesquiterpene lactone skeleton was provided.

Point 3: In figure 3, NOE interactions between H3 – H9 and H5 H9 are shown; however, the orientation observed for the proton 9 in the conformers included in this figure do not become possible this interaction. Are the conformers presented in this figure the lowest energy ones? It is necessary the authors clarified this situation.

Response 3: The minimize energy was generated and assaigned NOESY interactions were indicated according to the NOESY analysis. This is the only correlation for H-3 which correlate with H-9alpha. Additionally H-9alpha correlate H-1 and H-1 corrleated with H-5 which indicated the alpha position for H-3.

Point 4: The quality of figure 3 is not good. It is important to improve it.

Response 4: The quality of figure 3 has been improved

Point 5:  In the biosynthetic pathway discussion, the description of the compounds biotransformations to achieve product 6 is suitable, but the description of the biochemical pathways to produce compounds 1, 2, 5 and 4, actually is missing. The authors should include a brief description of these biotransformations.

Response 5: The biosynthetic pathway discussion of compounds 1,2,5 and 4 were briefly described.

Reviewer 3 Report

The paper concerns with the phytochemical studies about Centaurothamnus maximus aerial parts extract. The structural elucidation of two new guaianolides was realized using a full set of 1D and 2 NMR sequences after several steps of chromatography.

This work is not very innovative. This study developed conventional techniques for structural elucidation using 1D and 2D NMR sequences. Thus, the paper has a bit lack of the originality.

The technical aspects of the presented data seem to be good, and the analytical method is performed accurately. In the chemotaxoxomic significance section, the authors provide a long presentation of their results and the conclusions are accurate.

For these reasons, in my opinion the manuscript is suitable for the publication. Based in these results, the present paper should be accepted for publication in Molecules after revision, as follow:

  • Page l line 83-84. The 13C NMR chemical shifts of compound 1 and compound 7 (chlorine instead of hydroxyl) are very close. Please modify the sentence lines 84-87 including this comparison. Then, the 13C NMR chemical shifts of compounds 4 and 7 should be added in table 1 for an easy comparison;
  • Figure 1. Please add the numbering of carbons in molecule 1
  • Figure 4.The biosynthetic pathway should be simplified with costunolide as precursor.
  • In the same way, the biosynthetic pathway of figure 5 must start with chalcone;
  • The legend of figure 5 is not correct (methoxyled flavones instead of guaianolides). Please modify;
  • In the references several authors names are truncated (references 1, 2, 4,10, 11, 12;, 31, 42, 55, …). Please check all the references.

Author Response

Response to Reviewer 3 Comments

Thanks for respected reviewer for his comments to improve our manuscript.

Point 1:  Page l line 83-84. The 13C NMR chemical shifts of compound 1 and compound 7 (chlorine instead of hydroxyl) are very close. Please modify the sentence lines 84-87 including this comparison. Then, the 13C NMR chemical shifts of compounds 4 and 7 should be added in table 1 for an easy comparison;

Response 1: The sentence reveled the upfiled shift of C-15 (Cl) in compound 1 in comparison with compound 7 was added. Additionally, The 13C chemical shoft for compounds (3-7) were added to be easy comparison for reader

Point 2:  Figure 1. Please add the numbering of carbons in molecule 1

Response 2: the numbering of carbons in compound 1 and 2 were added

Point 3:  Figure 4.The biosynthetic pathway should be simplified with costunolide as precursor

Response 3: Figure 4 was simplified to be started with costunolide as precursor.

Point 4:  In the same way, the biosynthetic pathway of figure 5 must start with chalcone;

Response 4: Figure 5 was  simplified to be started with chalcone as precursor.

Point 5:  The legend of figure 5 is not correct (methoxyled flavones instead of guaianolides). Please modify;

Response 5: sorry for this type mistake, the figure legend was corrected

Point 6:  In the references several authors names are truncated (references 1, 2, 4,10, 11, 12;, 31, 42, 55, …). Please check all the references.

Response 6: Authors names were corrected

Round 2

Reviewer 3 Report

The manuscript has been deeply modified as suggested band has been significantly improved.

However, several typing minor errors are present mainly in References part.

Minor corrections

Line 92 e.g. italics

Reference 7. The authors are: Barrero, A. F.; Sánchez, J. F.;  Rodríguez, I.

Reference 35. Watson R.R.

Reference 36. Rangari V.D.

References 4, 38 and 66. Do not use capital letters

Reference 56. Stevens, K.L.

Reference 46. The authors are: Nowak G.; Dawid-Pać R.; Urbańska M.; Nawrot J.

References 50-53. Please homogenise the author name : González